# Anti-Inflammatory and Neuroprotective Effects of *Undaria pinnatifida* Fucoidan

**DOI:** 10.3390/md23090350

**Published:** 2025-08-29

**Authors:** Cheng Yang, Corinna Dwan, Barbara C. Wimmer, Sayed Koushik Ahamed, Fionnghuala James, Jigme Thinley, Richard Wilson, Luke Johnson, Vanni Caruso

**Affiliations:** 1Faculty of Health, School of Pharmacy and Pharmacology, University of Tasmania, Hobart, TAS 7005, Australia; cheng.yang@utas.edu.au (C.Y.); sayed.ahamed@utas.edu.au (S.K.A.); fionnghuala.james@utas.edu.au (F.J.); jthinley@utas.edu.au (J.T.); 2Marinova Pty Ltd., 249 Kennedy Drive, Cambridge, TAS 7170, Australia; corinna.dwan@marinova.com.au (C.D.); barbara.wimmer@marinova.com.au (B.C.W.); 3Central Science Laboratory, College of Science and Engineering, University of Tasmania, Hobart, TAS 7001, Australia; richard.wilson@utas.edu.au; 4School of Psychological Sciences, Psychology, University of Tasmania, Launceston, TAS 7248, Australia; lukejohnsonphd@gmail.com

**Keywords:** fucoidan, *Undaria pinnatifida*, anti-inflammatory agents, neuroprotective, oxidative stress, immune response

## Abstract

*Undaria pinnatifida* fucoidan (UPF), a sulphated polysaccharide derived from brown seaweed, has attracted scientific and clinical interest for its wide-ranging anti-inflammatory and neurodegenerative properties. A growing body of research shows that UPF inhibits NF-κB and MAPK signalling pathways, reduces pro-inflammatory cytokines (TNF-α, IL-1β, IL-6), decreases ROS production, and suppresses iNOS and COX-2 activity, thereby mitigating oxidative and inflammatory damage in vitro. In vivo studies confirm these actions, demonstrating reduced systemic inflammation, promoted antioxidant defence, modulated gut microbiota composition, and improved production of beneficial microbial metabolites. In parallel, emerging evidence highlights UPF’s neuroprotective potential, characterised by protection against neuroinflammation and oxidative stress, the attenuation of amyloid-beta deposition, and improvement in neuronal function. Importantly, low- to medium-molecular-weight and highly sulphated UPF fractions consistently exhibit stronger bioactivities, suggesting a structural basis for its therapeutic potential. This review integrates mechanistic evidence from cellular, preclinical, and emerging clinical studies, highlighting UPF as a versatile marine-derived agent with therapeutic relevance for inflammatory and neurodegenerative diseases, and outlines future research directions toward clinical translation.

## 1. Introduction

Fucoidans are a class of highly sulphated polysaccharides that are abundantly present in the cell walls of brown algae, including *Undaria pinnatifida* [1]. These polysaccharides have attracted considerable scientific attention due to their diverse and potent biological activities in the treatment of inflammatory-related diseases [2], metabolic disorders [3], cardiovascular conditions [4], and several cancers [5,6,7].

The structure of *Undaria pinnatifida* fucoidan (UPF) is characterised by its sulphated galactofucan architecture, where a fucose-galactose backbone with α-(1→3)-linked L-fucose is interspersed with α-(1→4)/α-(1→3)/α-(1→6) galactosyl residues that introduce branch points, as well as sulphate esters that are characteristically placed at C2/C4 of fucose and at C3/C4 of galactose [8,9]. In contrast, *Fucus* spp. (typical “type II” fucoidans) predominantly feature an all-fucose backbone with alternating α-(1→3)/α-(1→4) linkages and sulphate mainly at C2/C4, with far less galactose incorporation [10,11]. Across extracts, UPF commonly presents as moderately to highly sulphated material spanning a broad molecular-weight distribution, from low-molecular-weight (LMW) fractions (<10 kDa) to very high-molecular-weight (HMW) material (>300 kDa), with many food/biomedical fractions falling in the range of 10 to 500 kDa, largely dictated by harvest and processing [12,13]. Its co-monomer composition typically includes abundant galactose alongside fucose, with minor xylose, glucose, mannose, and occasional uronic acids, supporting the galactofucan designation and differentiating UPF from fucoidans of *Laminaria* or *Fucus* that are often more fucose-dominant [14,15,16,17]. Collectively, UPF’s C2/C4-focused sulphation, galactose-enriched backbone with branching via α-(1→6) galactose, and broad yet tuneable molecular-weight profile provide a clear structural context for interpreting its distinct bioactivity relative to other brown-algal fucoidans [8]. The structural diversity of fucoidans, shaped by factors such as seaweed species, degree and position of sulphation, molecular weight, and extraction methods, plays a crucial role in determining their bioactivity [18,19,20,21].

Among the various sources of fucoidan, recent research has increasingly focused on UPF, which has shown notable anti-inflammatory [22], antioxidant [23], immunomodulatory [24], antiviral [25], and neuroprotective effects [26], suggesting its significant potential in biomedical applications.

This review aims to provide a comprehensive overview of the anti-inflammatory and neuroprotective activities of UPF. Specifically, it synthesises current evidence on the molecular mechanisms responsible for these biological effects and evaluates the therapeutic potential of UPF in the management of chronic inflammatory conditions and neurodegenerative diseases, identifying directions for future research and clinical applications.

## 2. Materials and Methods

This literature review synthesises findings from peer-reviewed articles obtained through systematic searches of PubMed, Scopus, Web of Science, and ScienceDirect covering the period from 2000 to 2025. Keywords used included “*Undaria pinnatifida*”, “fucoidan,” “sulphated polysaccharide”, “structure,” “biological activity,” “anti-inflammation,” “neuroprotection,” and “mechanisms”. Only peer-reviewed original research articles and review papers published in English were considered. Inclusion criteria were studies that investigated the biological activities of UPF or its fractions in in vitro, in vivo, or clinical trials; provided mechanistic insights related to inflammation or neuroprotection; and reported clear experimental results. Exclusion criteria included conference abstracts, non-peer-reviewed reports, studies not specific to UPF, and those lacking structural, mechanistic, or biological relevance. Reference lists of selected articles were also screened to identify additional eligible publications. In total, 129 studies were included in this review.

For each included study, we extracted experimental settings and key findings of UPF used, including model/system, dose/regimen, main action mechanisms, and references in the main text. We also recorded physicochemical and sourcing details of UPF used, including algal part/tissue, origin (including supplier, if stated), extraction and purification, molecular weight, degree of sulphation, content of uronic acid, purity, monosaccharide composition (ratios of fucose, galactose, xylose, glucose, mannose, glucuronic acid, arabinose, rhamnose, fructose, and galacturonic acid), and references in the Appendix A. When a parameter was not reported in the source article, it was coded as NR (not reported).

## 3. Anti-Inflammatory Activity of UPF

### 3.1. In Vitro Studies

Several in vitro studies have indicated that UPF effectively inhibits the nuclear factor kappa B (NF-κB) and mitogen-activated protein kinase (MAPK) signalling pathways, suppresses oxidative stress, and reduces the expression of pro-inflammatory mediators, thereby mitigating diverse inflammation-related responses (Figure 1). 

A recent investigation has shown that a 4-h pre-treatment with UPF (10, 50, and 100 µg/mL) significantly suppressed the lipopolysaccharide (LPS)-induced upregulation of pro-inflammatory cytokines, including tumour necrosis factor alpha (TNF-α), interleukin-1 beta (IL-1β), and IL-6, in human macrophages and peripheral blood mononuclear cells (PBMCs) [27]. A low-molecular-weight UPF (100 µg/mL) also exhibited significant anti-inflammation effects by downregulating the gene expression of these pro-inflammatory cytokines in LPS-induced RAW264.7 cells [28]. Similarly, in a viral challenge model, UPF (200 µg/mL) reduced pro-inflammatory cytokines, including IL-6, IFN-α, interferon gamma (IFN-γ), and TNF-α, in SARS-CoV-2 infected Caco-2-N^int^ cells [25]. These effects are primarily mediated through inhibition of key inflammatory signalling pathways, such as nuclear factor kappa-light-chain-enhancer of activated B cells (NF-κB) and mitogen-activated protein kinase (MAPK), which are known to regulate cytokine gene transcription [29]. For example, treatment with UPF suppresses the nuclear translocation of NF-κB p65 and reduces the phosphorylation of p38 and extracellular signal-regulated kinases (ERK)1/2 MAPKs, leading to a marked decrease in pro-inflammatory cytokine expression [1,28,29]. Vaamonde-García et al. (2021) reported that UPF (5 µg/mL) treatment attenuated IL-1β-induced inflammation in osteoarthritic chondrocytes by blocking nuclear translocation of NF-κB and inhibiting its activation [30]. In addition, UPF inhibited NF-κB signalling and downregulated IL-6 and TNF-α in human colon carcinoma cell line (Caco-2) cells [31]. 

The suppression of oxidative stress represents a fundamental mechanism by which UPF alleviates inflammation. Oxidative stress, driven by excess reactive oxygen species (ROS), is a key initiator of inflammatory signalling cascades in various cell types [32]. In intestinal epithelial cell line (IEC-6) cells, UPF (100 µg/mL) significantly reduced oxidative damage induced by hydrogen peroxide (H_2_O_2_), as evidenced by decreased levels of malondialdehyde (MDA) and increased activity of antioxidant enzymes, including catalase (CAT), total superoxide dismutase (T-SOD), and glutathione (GSH) [33]. These changes were associated with protection against apoptosis and inhibition of pro-inflammatory responses, indicating that UPF enhances cellular antioxidant defences to maintain redox balance. Moreover, in RAW264.7 macrophages, low molecular weight UPF (100 µg/mL) inhibited LPS-induced ROS production and suppressed the phosphorylation of key MAPK signalling proteins (p38, ERK1/2, and JNK), leading to a significant reduction in the expression of inflammatory markers such as TNF-α, IL-6, and IL-1β [28]. Phull et al. (2017) found that UPF (15.52–500 µg/mL) exerted significant antioxidant activity in a dose-dependent manner in various in vitro antioxidant assays, including iron chelating, hydroxyl, nitric oxide, and DPPH activity, along with a reduction in inflammation responses in rabbit articular chondrocytes [34]. Another in vitro study also indicated that *Undaria pinnatifida* water extract (UPE) obtained by ultrasonication (200 and 400 µg/mL) significantly suppressed ROS production and restored H_2_O_2_-induced viability reduction in monkey kidney (Vero) cells in a dose-dependent manner. [35]. The cell-protective activity of the extract in this study was attributed to its capability to decrease pro-apoptotic protein (Bax) and increase anti-apoptotic protein (Bcl-2) [35].

Interestingly, there are an increasing number of studies correlating the antioxidant properties of UPF with its sulphate content and molecular weight [9,12,14,35,36,37]. For instance, UPF fractions with higher sulphation levels have been shown to exhibit significantly greater antioxidant activity compared to their lower-sulphated counterparts [9]. Moreover, fractionation studies revealed that low molecular weight UPF components possess enhanced antioxidant effects relative to high molecular weight forms, particularly in assays such as DPPH radical scavenging and ferric-reducing antioxidant power (FRAP) tests [12].

UPF inhibits major inflammatory mediators, including inducible nitric oxide synthase (iNOS), cyclooxygenase-2 (COX-2), nitric oxide (NO), and prostaglandin E2 (PGE2), whose overproduction exacerbates inflammatory responses and tissue damage [38,39]. For instance, low molecular weight UPF (100 µg/mL) alleviated LPS-induced inflammation in RAW264.7 cells by suppressing iNOS and COX2 activities [28]. Similarly, Song et al. in 2015 demonstrated that UPF (50 µg/mL) significantly inhibited iNOS and COX-2 expression, as well as attenuated the production of NO and PGE2 in LPS-stimulated RAW264.7 macrophages [40]. Lim et al. (2022) also reported that high molecular weight UPF (500 µg/mL) mitigated MG-H1-induced inflammation in Caco-2 cells by suppressing protein expression of COX-2 and iNOS and inhibiting NF-κB activation [31]. Additionally, UPF exerted potent anti-inflammatory effects in rabbit articular chondrocytes, where UPF significantly reduced COX-2 expression in a dose-dependent (0–100 µg/mL, 24 h) and time-dependent (30 µg/mL, 0–48 h) manner [34]. Moreover, Vaamonde-García et al. found that UPF (5 µg/mL) significantly inhibited IL-1β-induced production of NO, PGE2, and IL-6 in osteoarthritic chondrocytes, suggesting an immunomodulatory role of UPF in inflammatory conditions [30].

Recent in vitro investigations have suggested that UPF is capable of suppressing chemokine secretion in various inflammatory cells. Chemokines, also known as chemotactic cytokines, are a family of small signalling proteins that significantly contribute to regulating the migration and activation of immune cells during inflammatory responses [1,41]. According to the structure of N-terminal cysteine residues, chemokines are classified into four major subfamilies, including CXXXC (fractalkine), C-X-C (IL-8), C-C (monocyte chemoattractant protein [MCP-1], or monocyte inflammatory protein [MIP-1α], and MIP-1β), and C chemokines (lymphotactin) [41]. Chen et al. (2025) demonstrated that sulphated *Undaria pinnatifida* polysaccharides (50 and 200 µg/mL) significantly reduced MCP-1 production in vitro during oxalate crystal-induced inflammation in renal cells [42]. This reduction was linked to decreased cellular inflammation and oxidative stress, indicating the potential role of UPF in modulating chemokine-driven immune cell recruitment [42]. Kim et al. also reported that UPF treatment (100 µg/mL) significantly suppressed MCP-1 expression in 3T3-L1 adipocytes, indicating that UPF inhibits inflammation-associated chemokine signalling during adipocyte differentiation [43]. In addition, Vaamonde-García et al. (2022) indicated that UPF (5 µg/mL) significantly downregulated IL-6 and IL-8 (CXCL8) in IL-1β-induced human chondrocyte cells [44]. Similarly, Wimmer et al. (2025) demonstrated that UPF significantly reduced the secretion of pro-inflammatory chemokines (IL-8 and MCP-1) and increased production of anti-inflammatory cytokines (IL-6 and IL-10) in the Caco-2/THP-1 co-culture system after microbial stimulation [45], indicating that UPF can suppress immune cell recruitment and inflammatory signalling at the gut mucosal level. Moreover, a study on atopic dermatitis found that UPF significantly inhibited the mRNA expression of several key chemokines, including thymus- and activation-regulated chemokine (TARC), macrophage-derived chemokine (MDC), and RANTES (also known as CCL5), in TNF-α or IFN-γ-induced human epidermal keratinocytes [46]. 

The major outcomes derived from in vitro investigations into the anti-inflammatory effects of UPF are listed in Table 1.

### 3.2. In Vivo Studies

A substantial body of in vivo studies further supports the notion that UPF exerts its anti-inflammatory effects by suppressing pro-inflammatory cytokines such as TNF-α, IL-1β, and IL-6 [22,47,48,49]. Herath et al. (2020) reported that a 7-day oral administration of UPF (400 mg/kg/day) significantly attenuated particulate matter (PM) and ovalbumin (OVA)-induced IL-4, IL-17a, and IL-33 increase in lungs of a murine model of allergic airway inflammation [48]. In a later study, 27 days of UPF supplementation (400 mg/kg/day) significantly reduced TNF-α, IL-6, and IL-1β, and mitigated inflammation responses in the colon of dietary fibre deficiency (FD)-induced mice [33]. Similar effects of UPF were reported by Shi et al. (2024) in a Syrian hamster model of virus infection, where 6 days oral administration of UPF (200 mg/kg/day) alleviated SARS-CoV-2-induced lung and gastrointestinal tract injury by suppressing gene expression of TNF-α and IL-6 [25]. Lim et al. (2022) also indicated that 4 weeks oral administration of high molecular weight UPF (25 mg/kg/day) significantly inhibited MG-H1-caused TNF-α increase in mouse colon tissues [31]. Similarly, a 10-week oral administration of UPF (400 mg/kg/day) suppressed systemic inflammation in a high-fat diet (HFD)-induced obese mouse model [22]. The results of the study showed that UPF significantly reduced the expression of pro-inflammatory cytokines (TNF-α, IL-1β, and IL-6) in skeletal muscle, small intestine, and hypothalamus [22]. 

The antioxidant properties of UPF have been well demonstrated in animal models of inflammatory-related disorders. In HFD-fed mice, a 10-week oral administration of sulphated polysaccharides from *Undaria pinnatifida* significantly (100, 300, and 500 mg/kg/day) reduced markers of oxidative stress, including MDA and SOD in liver tissues, alongside an inhibition of triglycerides (TG), low-density lipoprotein cholesterol (LDL-c), and TNF-α production, suggesting that the suppression of oxidative stress contributed to hepatic lipid metabolism improvements, and mitigated HFD-induced inflammatory conditions [50]. Similarly, 7 days of UPF oral gavage (100 and 400 mg/kg/day) attenuated MDA in the serum and lungs of PM-induced allergic airway inflammatory mice [48]. Phull et al. (2017) also reported that 25 days UPF administration (150 mg/kg/day) significantly decreased arthritis-induced endogenous antioxidant enzymes such as CAT, peroxidase (POD), and SOD [34]. This reduction was mainly due to UPF capability to scavenge the free radicals, abrogate ROS-induced oxidative stress, and maintain the oxidative flux [34]. In contrast, Kang et al. found that 14 days UPF intraperitoneal administration (100 mg/kg/day) markedly prevented oxidative stress in carbon tetrachloride (CCL_4_)-induced rats by increasing antioxidant enzymes (CAT, SOD, and glutathione peroxidase [GPx]), and decreasing markers of oxidative damage (MDA) in liver [51]. These results are in line with the findings of Zheng et al. (2023), where 27 days of UPF treatment (400 mg/kg/day) significantly elevated the levels of CAT and T-SOD, and attenuated myeloperoxidase (MPO) and MDA production in the colon tissues of FD-induced mice [33], suggesting that UPF exerts a protective effect against inflammation-associated oxidative damage by enhancing endogenous antioxidant defences and reducing lipid peroxidation, thereby contributing to the amelioration of oxidative stress in various inflammatory disease models.

Recent studies have also demonstrated that UPF exhibits strong antioxidant activities in an in vivo zebrafish model, a vertebrate species with notable biochemical and physiological similarities to mammals [37,52]. The findings suggest that UPF effectively mitigates oxidative stress induced by 2,2′-azobis (2-amidinopropane) dihydrochloride (AAPH) and H_2_O_2_, as evidenced by increased survival rates, reduced cellular apoptosis, decreased heart rate, and lower levels of ROS and lipid peroxidation [37,52].

UPF has been shown to ameliorate inflammatory responses by modulating gut microbiota composition. A bidirectional relationship exists between gut dysbiosis and host inflammation, whereby microbial imbalance promotes inflammatory processes, which in turn further disrupt the gut microbial ecosystem [53,54,55,56]. Recent studies have indicated that UPF treatment significantly attenuated intestinal inflammation by restoring microbial balance, notably decreasing *Firmicutes* and increasing *Bacteroidetes* in the gut of HFD-induced obese mice [50,57,58,59,60]. Comparable outcomes were observed in FD-induced inflammatory mouse models, where 27 days of oral UPF administration (300 and 400 mg/kg/day) led to a marked increase in *Bacteroidetes* and a reduction in *Firmicutes* within colon tissues [33,61]. As the dominant phyla in the gut, *Firmicutes* and *Bacteroidetes* play key roles in maintaining intestinal homeostasis [62], and imbalances in their ratio have been associated with various inflammatory disorders [63,64,65,66]. The anti-inflammatory effects of UPF may also derive from its prebiotic properties, as evidenced by reductions in pathogenic taxa (*Faecalibaculum*, *Desulfovibrionales*, *Proteobacteria*, and *Clostridia*) and enrichment of beneficial bacteria (*Akkermania muciniphila*, *Bacteroides*, *Bifidobacterium spp.*, and *Lactobacillus*) [58,59,60,67]. Additionally, Park et al. (2024) reported that 4 weeks of UPF supplementation (50, 100, and 200 mg/kg/day) significantly increased the abundance of *Papillibacter cinnamivorans*, a butyrate-producing bacterium, in immunosuppressed rats [68]. Butyrate, one of the short-chain fatty acids (SCFAs), mitigates inflammation by interacting with immune cells, promoting anti-inflammatory cytokines, and suppressing pro-inflammatory mediators through G-protein coupled receptors (GPR41/43) and inhibition of histone deacetylases (HDACs) [69,70]. Similarly, Zheng et al. (2023) have suggested that 27 days of UPF supplementation (400 mg/kg/day) significantly restored HFD-induced reduction in colonic SCFAs, including acetate, propionate, and butyrate [33], suggesting that UPF may exert its anti-inflammatory effects, at least in part, by restoring SCFA levels and modulating immune responses through established SCFA-mediated pathways.

UPF has been reported to attenuate immune cell infiltration, including macrophages and T cells, and to ameliorate inflammatory responses in allergic conditions. Herath et al. (2020) indicated that 7 days of UPF oral gavage (400 mg/kg/day) significantly reduced PM-exacerbated infiltration of inflammatory cells, such as F4/80^+^ macrophages, CD4^+^ T lymphocytes, Gr-1^+^ granulocytes, and eosinophils, in the trachea and lungs of OVA-sensitised mice [48]. The results also showed that UPF decreased serum level of immunoglobulin E (IgE) and suppressed inflammatory provocation-induced increase in goblet cell hyperplasia and mucus secretion [48], suggesting potent therapeutic effects of UPF in allergic airway inflammation. Similarly, Yu et al. (2024) demonstrated that 16 days administration of ethanol-extracted UPE (50, 100, and 200 mg/kg/day) mitigated combined allergic rhinitis and asthma syndrome by inhibiting the accumulation of inflammatory cells, including epithelial cells, eosinophils, neutrophils, lymphocytes, and macrophages, in both nasal and bronchoalveolar lavage fluid, as well as a reduction in Th2 cytokines expression (IL-4, IL-5, and IL-13) [71].

The capability of UPF to re-establish immune homeostasis also plays a significant role in mitigating inflammatory conditions. Several in vivo studies have demonstrated that UPF exerts immunomodulatory effects by upregulating the expression of the anti-inflammatory cytokine IL-10 while concurrently downregulating the production of pro-inflammatory cytokines in various animal models of inflammatory intestinal diseases [50,59,61,72]. The main results of in vivo evaluations of UPF effects are listed in Table 2.

To aid interpretation, the study-level characteristics of the UPF used in each experiment (source, extraction/purification, molecular weight, sulphation, uronic acid, purity, monosaccharides) are summarised in Appendix A. For transparency, non-reported items are indicated as NR.

### 3.3. Clinical Trials

A growing body of clinical evidence supports the therapeutic potential of UPF in modulating metabolic, immune, and inflammatory responses in humans. Various human studies have demonstrated that incorporating *Undaria pinnatifida* (4–6 g/day) into diets help improve metabolic parameters, including suppressed postprandial glycaemia, modulated appetite sensations, reduced waist circumference and blood pressure, as well as a decrease in total cholesterol, LDL-cholesterol, and resistin levels [73,74,75,76]. *Undaria pinnatifida* also exerts potent immunostimulatory properties to manage Herpes infections by promoting healing and preventing reactivation [77]. Moreover, a clinical trial showed that a single dose of UPF (1 g) modulated microRNA expression related to immune response and inflammation, highlighting its systemic regulatory potential [78]. Additionally, a combination of UPF and *Fucus vesiculosus* fucoidan significantly increased faecal lysozyme levels, a protein known for its antimicrobial and anti-inflammatory functions, suggesting that UPF may promote mucosal barrier integrity and reduce mucosal inflammation [79]. Cox et al. (2023) reported that 3 weeks of UPF supplementation (1 g/day) significantly increased salivary immunoglobulin (Ig) A contents after an intensified training, indicating UPF properties to enhance mucosal immunity and provide protective anti-inflammatory benefits [80,81]. In a double-blind randomised placebo-controlled clinical trial, 2 weeks of UPF administration (1 g/day) significantly suppressed the upregulation of inflammatory cytokines induced by high-intensity exercise [82]. Similarly, UPF combined with green-lipped mussel mitigated joint pain and prediabetes in a randomised, double-blinded clinical setting, demonstrating that UPF elicits antioxidant and anti-inflammatory effects [83].

## 4. Neuroprotective Effect of UPF

The scientific literature has reported an increasing number of studies highlighting the neuroprotective effects of UPF in promoting brain health and mitigating the progression of neurodegenerative diseases such as Alzheimer’s (AD). These effects arise from multifactorial interplay involving anti-inflammatory [22], anti-apoptotic [84], antiviral [26], and antioxidant activities [85] of UPF. Findings from both in vitro and in vivo investigations demonstrate that UPF can attenuate neuronal apoptosis, inhibit amyloid-β (Aβ) aggregation, and suppress the activation of microglia and astrocytes by reducing oxidative stress and neuroinflammation across various experimental models of neurodegenerative conditions [22,26].

### 4.1. In Vitro Studies

Several in vitro studies demonstrated that UPF is capable of suppressing inflammation in neurodegenerative conditions, primarily through inhibition of the NF-κB signalling pathway, mitigation of oxidative stress, and modulation of adenosine monophosphate-activated protein kinase (AMPK) and mechanistic target of rapamycin (mTOR) pathways (Figure 2). 

Notably, Giuliani et al. (2025) reported that UPF (100 µg/mL) significantly attenuated herpes simplex virus type I (HSV-1)-induced AD-like pathology [26]. This included a reduction in amyloid precursor protein (APP) production and Aβ synthesis, alongside inhibiting NF-κB pathway activation and reducing IL-6 expression [26]. Ethanol-extracted UPE (5 µg/mL) also has been suggested to reduce endoplasmic reticulum (ER) stress and increase cell viability in hypothalamic neurons via Akt/mTOR signalling, highlighting its anti-inflammatory and neuroprotective potential [86]. ER stress is closely linked to the activation of inflammatory responses and is increasingly recognised as a contributing factor in the pathogenesis of various neurodegenerative diseases [87]. Additionally, Chen et al. (2025) suggested that low molecular weight UPF (0.125 mg/mL) significantly ameliorated LPS-induced macrophage inflammatory state by promoting its polarisation from pro-inflammatory M1 phenotype to anti-inflammatory M2 phenotype through the AMPK/mTOR pathway [88]. Literature has indicated that modulating the AMPK/mTOR pathway regulates microglia polarisation and reduces neuroinflammation [89].

UPF has also been shown to exert neuroprotective effects in various in vitro cell models of neurodegeneration by enhancing cell viability and attenuating cytotoxicity, particularly in response to neurotoxic insults such as Aβ and oxidative stress [26,84,85,90,91,92]. For instance, Wei et al. (2017) demonstrated that pre-treatment with UPF (100, 200, 400 µg/mL) for 24 h protected PC12 cells from apoptosis induced by Aβ_25–35_ and d-galactose (D-Gal), alongside elevated levels of SOD and GSH [85]. Similar effects were observed in a rat cholinergic basal forebrain neuron model of AD conditions, where treatment of a commercial UPF (1 µM) inhibited cellular and neurotoxic effects of Aβ_1−42_ and suppressed ROS production [90]. In addition, UPF demonstrated strong free radical-scavenging activity, effectively inhibiting DPPH and hydroxyl radicals, and reducing ROS production as well as Aβ synthesis in HSV-1-infected retinal pigment epithelium (RPE) cells [26]. HSV-1 infection and Aβ synthesis have been associated with the development of AD [93]. Mohibbullah et al. (2018) also reported that ethanol-extracted UPE (15 µg/mL) enhanced cell viability and reduced cytotoxicity in hippocampal neurons by decreasing ROS generation, membrane phosphatidylserine exposure, genomic DNA degradation, and restoring hypoxia-induced mitochondrial depolarization [92]. Notably, although both fucoidans reduced Aβ_1–42_-induced oxidative stress and apoptosis levels, UPF exhibited stronger neuroprotective effects than *Fucus vesiculosus* fucoidan, likely due to its distinct structural features such as higher sulphate content and specific molecular weight distribution [84]. 

The key in vitro outcomes regarding the neuroprotective activity of UPF are comprehensively outlined in Table 3.

### 4.2. In Vivo Studies

UPF has emerged as a promising neuroprotective agent due to its ability to attenuate neuroinflammation. For example, 10 weeks of UPF oral administration (400 mg/kg/day) significantly attenuated HFD-induced neuroinflammation in obese mice by downregulating the expression of pro-inflammatory cytokines (TNF-α, IL-1β, IL-6, and IFN-γ) in hypothalamic tissues and suppressing the production of inflammation-related proteins (leucine-rich repeat serine/threonine-protein kinase 2 [Lrrk2], wolframin [Wfs1], and neuroglobin [Ngb]) in the nucleus accumbens [22]. Hu et al. (2014) also reported that a 10-day intrathecal injection of commercial UPF (15, 50, and 100 mg/kg/day) mitigated rat neuropathic pain induced by L5 spinal nerve ligation (SNL). The findings suggested that UPF inhibited microglia and astrocyte activation in the lumbar spinal cord and reduced TNF-α, IL-1β, and IL-6 expression in the spinal dorsal horn [94]. Similarly, Che et al. (2017) demonstrated that intraperitoneal injection of commercial UPF (80 and 160 mg/kg/day) for 7 days significantly ameliorated cerebral ischemia–reperfusion injury (IRI)-caused neurological impairment in rats and significantly decreased the levels of pro-inflammatory cytokines, including IL-1β, IL-6, MPO, and TNF-α [95].

Several in vivo studies have highlighted the effects of UPF in neuroprotection and rehabilitation, demonstrating its beneficial properties in inhibiting oxidative stress and attenuating neurotoxic protein aggregation. Specifically, 21 days of oral administration of UPF (50, 100, and 200 mg/kg/day) improved learning and memory impairments in AD-model mice induced by D-Gal, where UPF exhibited potent antioxidant effects, enhancing SOD and GSH activity [85]. The ability of UPF to promote learning and memory in this study is largely attributed to its enhancement of acetylcholine (ACh) content and choline acetyl transferase (ChAT) enzyme activity, along with the inhibition of acetylcholine esterase (AChE) enzyme activity, which are key factors involved in the cognitive dysfunction characteristic of AD [85]. UPF also has been reported to reduce oxidative stress-related proteins (SOD and MDA), suppress pro-apoptotic proteins (p-p53 and Bax), and elevate anti-apoptotic protein (Bcl-2) in IRI-induced rats by inhibiting MAPK pathway [95]. Similarly, Wang et al. (2016) illustrated that intraperitoneal pre-treatment of low molecular weight commercial UPF (50 mg/kg) significantly suppressed neuronal damage and neurological deficits in aged mice after traumatic brain injury (TBI), where UPF exerted these protective effects by inhibiting oxidative stress (reduced MDA, 4-hydroxynonenal [4-HNE], ROS and increased CAT, SOD, GPx) and mitochondrial dysfunction (suppressed cytochrome c release) [96]. In addition, the neuroprotective effects of UPF have been found in an invertebrate model of AD, where UPF (500 ng/mL) alleviated Aβ-induced paralysis by decreasing Aβ deposition and ROS production in transgenic Caenorhabditis elegans [97]. Taken together, these findings suggest that UPF confers neuroprotection across diverse experimental models by modulating oxidative stress, mitochondrial integrity, and apoptosis-related pathways, supporting its potential as a therapeutic agent in the prevention and treatment of neurodegenerative disorders (Table 4).

Key physicochemical information for the UPF used in the cited studies is provided in Appendix A (source, extraction/purification, molecular weight, sulphation, uronic acid, purity, monosaccharides), with unreported parameters marked as NR.

## 5. Discussion

### 5.1. Toxicity Study of UPF

UPF demonstrates a favourable safety profile across preclinical and clinical studies. The U.S. Food and Drug Administration (FDA) issued “no questions” responses to GRAS notices for fucoidans derived from *Undaria pinnatifida* and *Fucus vesiculosus*, supporting their safety for use in food applications with estimated daily intakes of up to 250 mg/day [98,99]. In animal models, repeated oral administration of UPF for four weeks at doses up to 1350 mg/kg/day in SD rats produced no treatment-related adverse effects on clinical signs, body weight, haematology, or histopathology [100]. Similarly, sub-chronic assessment of fucoidan from *Laminaria japonica* administered to Wistar rats for six months demonstrated no significant toxicological effects at 300 mg/kg/day [101]. Additionally, standard genotoxicity tests, including the bacterial reverse-mutation assay, in vitro chromosomal aberration, and in vivo micronucleus test, demonstrated no mutagenic or clastogenic potential [102]. Human data further support the safety of UPF use. A pilot study administering 3 g/day of UPF for 12 days reported mild, subclinical prolongation of activated partial thromboplastin time, without clinically significant anticoagulant activity [103]. Other trials with oral doses of 1–3 g/day for several weeks to months reported no adverse effects or alterations in liver and kidney function [98,99]. Given fucoidan’s structural similarity to heparin, theoretical anticoagulant interactions remain a consideration, particularly in patients on antithrombotic therapy [104,105]. Importantly, safety is influenced by extraction and purification. Commercial UPF preparations, such as those produced using proprietary aqueous extraction processes, are highly purified and standardised, ensuring consistency and minimising the risk of contaminants [45,49]. Collectively, UPF appears non-toxic, non-genotoxic, and well-tolerated in both preclinical and clinical settings, supporting its safe use as a functional ingredient and potential therapeutic agent, with caution advised for individuals with bleeding disorders or on anticoagulant therapy.

### 5.2. Pharmacokinetics of UPF

The pharmacokinetics of UPF have not yet been comprehensively characterised, representing an important gap in linking systemic exposure with its biological effects. Limited in vitro work has shown that UPF interacts minimally with hepatic CYP450 enzymes but may influence catechol-O-methyltransferase metabolism [106]. However, studies of fucoidans from other brown algae suggest limited gastrointestinal absorption, influenced by molecular weight, sulfation degree, and structural composition [107,108,109]. For instance, orally administered fucoidan from *Fucus vesiculosus* in rats showed prolonged blood circulation (mean residence time = 6.79 h) and preferential accumulation in the kidneys, spleen, and liver, with relatively low absorption due to high molecular weight and sulphation [109]. Similarly, in mice, low-molecular-weight fucoidan (9.5 kDa) from *Saccharina japonica* exhibited more rapid absorption (reached maximum concentration at 1.5 h) and higher bioavailability (28.3 %) compared to its high-molecular-weight counterpart [110]. Mechanistic studies using fluorescein-isothiocyanate-labelled fucoidan indicate clathrin-mediated intestinal uptake and tissue distribution, notably to the kidney and liver [111]. Additionally, a Heparin Red fluorescence assay has been shown to enable direct quantification of UPF and other fucoidans in human plasma (0.5–20 µg/mL) [112]. Other human trials have also detected fucoidans (from *Undaria pinnatifida* and *Cladosiphon okamuranus*) can cross the intestinal wall and enter circulation, with absorption efficiency estimated at up to 1% of the oral dose absorbed [113,114]. Taken together, these findings suggest that while systemic uptake of oral fucoidan is limited, it is measurable and may be sufficient to exert biological effects directly or indirectly through gastrointestinal interactions and the generation of bioactive oligosaccharides. 

By contrast, the high molecular weight and polyanionic characteristics of UPF favour skin-surface and epidermal retention with limited systemic exposure after topical use [115]. For example, in a mouse barrier-disruption model, UPF (171 kDa) accelerated transepidermal water loss recovery and normalised keratinocyte differentiation markers, consistent with effective local bioavailability in the viable epidermis [115]. However, although topical pharmacokinetics of UPF remain largely unexplored, a closely related fucoidan ointment (735 kDa, source *Fucus vesiculosus*) showed linear pharmacokinetics after single topical doses (50–150 mg/kg) in rats, with sustained plasma profiles, an apparent elimination half-life of 20.75 ± 9.43 h, and low absolute bioavailability (17.7 ± 7.7% of the applied dose) [116]. Tissue measurements demonstrated a pronounced skin “reservoir” with measurable permeation to underlying muscle but no plasma accumulation after 5 once-daily doses (100 mg/kg), supporting predominantly local disposition with limited systemic exposure [116]. Although derived from *Fucus vesiculosus*, these findings are mechanistically applicable to UPF given the shared high molecular weight and sulphate density that govern dermal transport. Topical fucoidan (*Cladosiphon okamuranus*, 49.8 kDa) has also demonstrated anti-inflammatory activity in the skin, as seen in atopic dermatitis and wound models, including reductions in mast cells, epidermal thickness, and serum IgE in mice, as well as broader dermatologic benefits in human cosmetic testing [117,118,119]. Although systemic exposure to topically applied fucoidan is low, local dermal exposure yields significant pharmacological effects [116,120]. For instance, a *Fucus vesiculosus* fucoidan (735 kDa) cream dose-dependently inhibited carrageenan-induced oedema and mechanical allodynia in rats, with high-dose efficacy comparable to diclofenac gel [120]. These efficacy data align with pharmacokinetics findings of predominant local tissue disposition after topical dosing [120]. Importantly, the lack of comprehensive pharmacokinetic studies on UPF represents a key knowledge gap, and future work is needed to establish dose-exposure-response relationships that can better explain its anti-inflammatory and neuroprotective activities.

### 5.3. Structure–Activity Relationships of UPF

Synthesis of the available data reveals recurring trends across the reviewed studies. In terms of molecular weight, low- to medium-molecular-weight UPF often shows stronger anti-inflammatory readouts, such as reduced NF-κB/MAPK signalling, iNOS/COX-2, and pro-inflammatory cytokines [27,28,30,42,44,47,71,88]. Fractionation studies also report superior primary antioxidant capacity for <10 kDa compared to >300 kDa fractions, consistent with better diffusion/receptor access [12]. By contrast, very high-molecular-weight UPF can excel in viscosity/barrier-related endpoints, restoring epithelial integrity in MG-H1 injury models in Caco-2 cells and mice [31]. Regarding to the degree of sulphation, UPF is typically a sulphated galactofucan with sulphate esters at fucose C2/C4 and galactose C3/C4 [8,9], and higher sulphate density in fucoidans generally strengthens protein–polysaccharide interactions [121,122], such as P-selectin binding, and downstream anti-inflammatory effects, though activity remains chemotype- and context-dependent [123]. Additionally, for co-monomer composition/branching, *Undaria pinnatifida* commonly bears substantial galactose (galactofucan) with branching (including 1,6-linked galactose) [124,125], and this architecture recurs in UPF studies reporting mucosal and immunomodulatory benefits, for instance, butyrogenic shifts and cytokine dampening in Simulator of the Human Intestinal Microbial Ecosystem (SHIME) and mouse gut models [33,45,81,126], and improved mucosal immune markers in athletes [79], whereas more fucose-dominant chemotypes from other algae, such as *Fucus vesiculosus*, can differ in neuroprotective potencies/selectivity [84]. A current literature also reported, compared to other species, a purified *Fucus vesiculosus* fucoidan (high fucose, reduced phenolics) inhibited protein denaturation with IC_50_ of 0.20 mg/mL, outperforming diclofenac (0.37 mg/mL), and stabilised human red blood corpuscles (HRBC) membranes to 88% at 0.5 mg/mL underscoring strong species- and structure-dependence of these surrogate anti-inflammatory readouts [127].

### 5.4. Limitation

Several clinical studies (Section 3.3 Clinical trials) have reported promising anti-inflammatory and immunomodulatory effects of UPF, including improvements in metabolic parameters, mucosal immunity, and exercise-induced inflammation. However, its neuroprotective potential, although strongly supported by preclinical studies (Section 4. Neuroprotective Effects of UPF), has not yet been evaluated in human trials, suggesting the urgent need for well-designed, large-scale clinical studies to justify its therapeutic relevance in clinical practice. Additionally, several studies did not report one or more critical UPF characteristics, such as molecular weight, sulphate content, or monosaccharide composition. This suggests that cross-study comparisons should be interpreted cautiously. We therefore indicated non-reported items as NR in Appendix A in this review. Standardised reporting of UPF chemotype (source/tissue, extraction/purification, molecular weight, sulfation, and monosaccharides) would materially strengthen future syntheses and help relate structure to function.

## 6. Conclusions

*Undaria pinnatifida* fucoidan (UPF) represents a structurally unique sulphated galactofucan with anti-inflammatory and neuroprotective activities demonstrated in vitro, in vivo, and in ongoing clinical studies. Its anti-inflammatory effects are consistently mediated through inhibition of NF-κB and MAPK pathways, downregulation of pro-inflammatory cytokines and mediators, reduction in oxidative stress, and modulation of immune and gut microbiota responses. Neuroprotective actions observed in preclinical studies include attenuation of neuroinflammation, reduction in oxidative damage and amyloid burden, and enhancement of neuronal survival and antioxidant defences. Notably, low- to medium-molecular-weight and highly sulphated fractions exhibit stronger bioactivities, underscoring clear structure-activity relationships. Despite these promising findings, the molecular mechanisms underlying UPF’s actions remain incompletely understood, and its therapeutic effects in humans have yet to be fully confirmed. Future research should focus on clarifying molecular mechanisms, standardising UPF extraction and structural characterisation, and developing delivery strategies to optimise bioavailability. Most critically, rigorously designed clinical trials are required to confirm efficacy and safety, thereby supporting the integration of UPF into evidence-based therapeutic applications.

## Figures and Tables

**Figure 1 marinedrugs-23-00350-f001:**
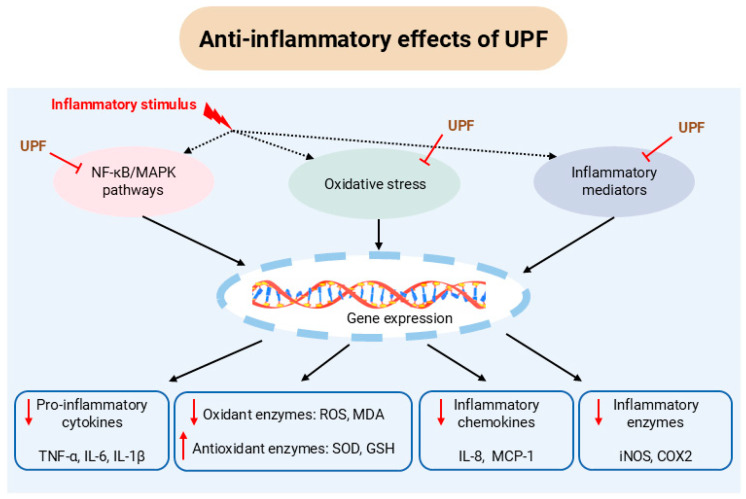
Anti-inflammatory activities of UPF in vitro. NF-κB: nuclear factor kappa B; MAPK: mitogen-activated protein kinase; TNF-α: tumour necrosis factor alpha; IL-6: interleukin-6; IL-1β: interleukin-1 beta; ROS: reactive oxygen species; MDA: malondialdehyde; SOD: superoxide dismutase; GSH: glutathione; IL-8: interleukin-8; MCP-1: monocyte chemoattractant protein-1; iNOS: inducible nitric oxide synthase; COX2: cyclooxygenase-2; UPF: *Undaria pinnatifida* fucoidan.

**Figure 2 marinedrugs-23-00350-f002:**
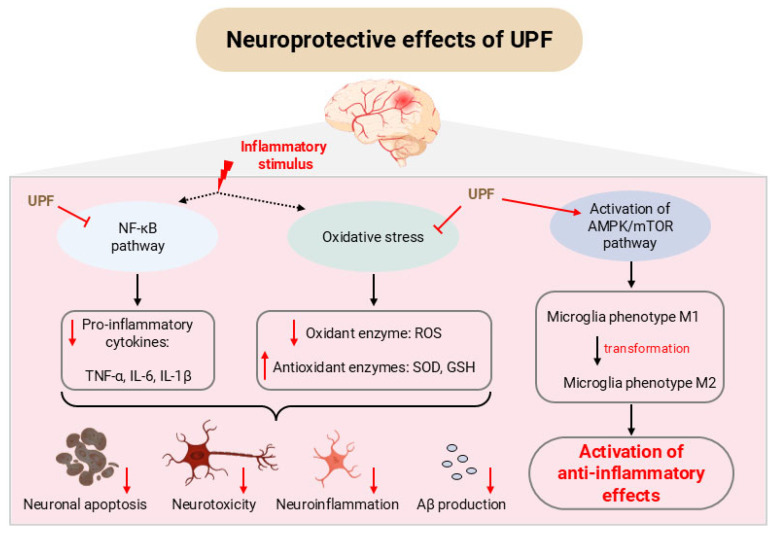
Neuroprotective effects of UPF in vitro (with the use of Biovisart.com.cn, accessed on 19 August 2025). NF-κB: nuclear factor kappa B; TNF-α: tumour necrosis factor alpha; IL-6: interleukin-6; IL-1β: interleukin-1 beta; ROS: reactive oxygen species; SOD: superoxide dismutase; GSH: glutathione; Aβ: amyloid-beta peptides; AMPK: adenosine monophosphate-activated protein kinase; mTOR: mechanistic target of rapamycin; UPF: *Undaria pinnatifida* fucoidan.

**Table 1 marinedrugs-23-00350-t001:** In vitro anti-inflammatory activity of UPF.

Cell Line	Model	Compound	Tested Concentration	Effective Concentration	Activity	Reference
Murine RAW264.7 cells	LPS stimulation	Low molecular weight UPF	1, 10 and 100 µg/mL	1, 10 and 100 µg/mL	Reduced ROS, COX-2 and iNOS; suppressed phosphorylation of p38, ERK1/2, and JNK; and downregulated TNF-α, IL-6, and IL-1β	[28]
		UPF	12.5, 25, and 50 µg/mL	50 µg/mL	Inhibited COX-2 and iNOS, and attenuated production of NO and PGE2	[40]
Human osteoarthritic chondrocytes	IL-1β- induced inflammation	UPF	5, 30, and 100 µg/mL	5 µg/mL	Inhibited NF-κB activation; and reduced production of NO, PGE2, and IL-6	[30]
		UPF	1, 5, and 30 µg/mL	5 µg/mL	Downregulated IL-6 and IL-8 (CXCL8); upregulated Nrf-2, HO-1, and SOD-2.	[44]
THP-1 cells and PBMCs	LPS-induced inflammation	UPF	10, 50, 100, and 200 µg/mL	10, 50, and 100 µg/mL	Reduced expression of TNF-α, IL-1β, and IL-6	[27]
Caco-2-N^int^ cells	SARS-CoV-2 infection	UPF	0–1000 µg/mL	200 µg/mL	Reduced IL-6, IFN-α, IFN-γ, and TNF-α	[25]
Caco-2 cells	MG-H1-induced inflammation	High molecular weight UPF	0–1000 µg/mL	500 µg/mL	Inhibited NF-κB signalling; downregulated IL-6 and TNF-α; and suppressed COX-2 and iNOS expression	[31]
IEC-6 cells	H_2_O_2_-induced oxidative stress	UPF	10, 20, 50, and 100 µg/mL	100 µg/mL	Decreased levels of MDA, and increased CAT, T-SOD, and GSH	[33]
Rabbit articular chondrocytes	Antioxidant assays	UPF	0–500 µg/mL	2.5–100 µg/mL	Reduced COX-2; scavenged DPPH, nitric oxide and hydroxyl radicals; and exhibited iron chelating activity	[34]
Vero cells	H_2_O_2_-induced viability reduction	Water-ultrasonicated UPE	50, 100, 200, and 400 µg/mL	200 and 400 µg/mL	Suppressed ROS production; decreased Bax; and increased Bcl-2	[35]
Human renal cells	Oxalate crystal-induced inflammation	sulphated *Undaria pinnatifida* polysaccharides	50, 100, 150, 200, and 250 µg/mL	200 µg/mL	Reduced ROS and MCP-1 production; increased SOD content; and decreased secretion of TNF-α and IL-1β	[42]
3T3-L1 adipocytes	Adipogenesis	UPF	1, 10, and 100 µg/mL	100 µg/mL	Reduced production of ROS, SOD, and GPx; and downregulated expression of TNF-α, MCP-1 and PAI-1	[43]
Caco-2/THP-1 coculture	Microbial stimulation	UPF	2.5 g/L	2.5 g/L	Reduced secretion of IL-8 and MCP-1; decreased TNF-α; and increased IL-6 and IL-10	[45]
Human epidermal keratinocyte cell line	TNF-α or IFN-γ-induced inflammation	UPF	400 µg/mL	400 µg/mL	Inhibited expression of TARC, MDC, and RANTES (CCL5)	[46]

**Table 2 marinedrugs-23-00350-t002:** In vivo anti-inflammatory activity of UPF.

Model	Animal	Compound	Dose	Treatment	Tissue	Result	Reference
HFD-induced obesity	Male and female C57BL/6J mice	UPF	400 mg/kg/day	Oral supplementation for 10 weeks	Skeletal Muscle	Reduced TNF-α, IL-1β, and IL-6	[22]
					Small Intestine	Reduced TNF-α, IL-1β, IL-6, NF-κB, Tjp1, GPR41, and GPR43	
					Plasma	Reduced IL-1α and IL-6	
	Male and female C57BL/6J mice	UPF	400 mg/kg/day	Oral supplementation for 10 weeks	Faeces	Increased abundance of *Bacteroidetes*, *Bacteroides/Prevotella*, *Akkermansia muciniphila*, and *Lactobacillus*; and reduced F/B ratio	[60]
	Male BALB/c mice	Sulphated polysaccharides from *Undaria pinnatifida*	150 and 300 mg/kg/day	Oral gavage for 10 weeks	Serum	Reduced levels of TC, TG, and LDL-c; increased HDL-c; suppressed FITC and LPS	[59]
					Liver	Increased expression of ABCG8, PPAR-γ, PGC-1α and CAT; reduced content of TC, TG, and MDA; and inhibited LPS production	
					Colon	Increased IL-10 expression; and reduced IL-6	
					Faeces	Increased abundance of *Bacteroidetes*, *Bacteroidaceae*, and *Prevotellaceae*; decreased *Firmicutes*, and *Proteobacteria*; increased levels acetate, propionate, and butyrate; and reduced F/B ratio	
HFD-induced obesity	Male BALB/c mice	Sulphated polysaccharides from *Undaria pinnatifida*	100, 300, and 500 mg/kg/day	Oral gavage for 10 weeks	Serum	Reduced levels of TC, TG, LDL-c, LPS, and FITC; and increased HDL-c	[50]
					Liver	Suppressed levels of LDL-c and MDA; and increased SOD	
					Colon	Decreased TNF-α; and increased IL-10	
					Faeces	Increased *Bacteroidetes* abundance; reduced *Firmicutes*, *Desulfovibrionales*, and *Clostridia*; and increased levels acetate, propionate, and butyrate	
	Male C57BL/6J mice	*Undaria pinnatifida* powder	10% (*w*/*w*)	Oral supplementation for 10 weeks	Faeces	Increased acetic acid, propionic acid, and butyric acid; increased *Bacteroidetes*, *Bacteroidaceae*, and *Bacteroides*; and reduced *Firmicutes*, *Lachnospiraceae*, *Streptococcaceae*, *Marinifilaceae*	[58]
HFD-induced dyslipidaemia	Male BALB/c mice	UPF	50 and 100 mg/kg/day	Oral gavage for 8 weeks	Serum	Suppressed levels of TC and LDL-c	[57]
					Liver	Attenuated levels of TG and CHO	
					Faeces	Increased *Bacteroidetes*; and reduced *Firmicutes*	
l-NAME-induced hypertension	Male SD rats	UPF	20 and 100 mg/kg/day	Oral gavage for 4 weeks	Thoracic aorta	Increased phosphorylation of eNOS and Akt; and decreased levels of iNOS and NO	[47]
					Serum	Decreased levels of TNF-α and IL-1β	
Particulate-matter-induced allergic airway inflammation	Female BALB/c mice	UPF	100 and 400 mg/kg/day	Oral gavage for 7 days	Lung	Suppressed MDA level; attenuated eosinophils, Gr-1+ cells, F4/80^+^ macrophage, and CD4^+^ T cell infiltration; and reduced IL-4, IL-17a, and IL-33	[48]
					Trachea	Attenuated eosinophils, Gr-1+ cells, F4/80^+^ macrophage, and CD4^+^ T cell infiltration	
					Serum	Inhibited MDA level; attenuated total IgE; and reduced IL-4	
Testosterone-induced BPH	Male SD rats	UPF	40 and 400 mg/kg/day	Oral administration for 4 weeks	Prostate	Reduced levels of testosterone and DHT; increased Bax; and reduced Bcl-2 expression	[49]
					Serum	Decreased levels of IL-1β, TNF-α, testosterone, DHT, and PSA	
Fibre deficiency-induced intestinal inflammation	Male BALB/c mice	UPF	100 and 400 mg/kg/day	Oral supplementation for 4 weeks	Colon	Increased levels of occludin, ZO-1, and claudin-3; reduced expression of TNF-α, IL-6, and IL-1β; increased IL-10; suppressed MDA, MPO, and LPS; promoted CAT and T-SOD; and increased production of acetate, propionate, and butyrate	[33]
	Male BALB/c mice	UPF	300 mg/kg/day	Oral gavage for 4 weeks	Colon	Reduced expression of TNF-α and IL-1β; elevated occludin and IL-10; increased levels of T-SOD and CAT; and decreased COX-2, iNOS, and LPS	[61]
					Faeces	Increased abundance of *Bacteroidetes* and *Bacteroidales*; and decreased *Firmicutes*, *Clostridiales*, and *Ruminococcaceae*	
SARS-CoV-2 infection	Female Syrian hamsters	UPF	100 and 200 mg/kg/day	Oral administration for 6 days	Lung	Downregulated ACE2, IL-6, and TNF-α	[25]
					Colon	Reduced levels of ACE2, IL-6, and TNF-α	
					Faeces	Decreased *Firmicutes*, *Limosillactobacter*; increased *Bacteroidota*, *Patescibacteria*, *Allobaculum*, *Candidatus Saccharimonas*, and *Ileibacteria*; and increased levels of acetate and propionate	
MG-H1-induced intestinal inflammation	Male ICR mice	High molecular weight UPF	25 and 75 mg/kg/day	Oral administration for 4 weeks	Colon	Inhibited MPO activity; and decreased expression of ZO-1, RAGE, and TNF-α	[31]
Carrageenan induced inflammation	Male SD rats	UPF	50 and 150 mg/kg/day	Oral gavage for 25 days	Serum	Decreased production of CAT, POD, and SOD	[34]
CCL4-induced oxidative stress	Female SD rats	UPF	100 mg/kg/day	Intraperitoneal injection for 2 weeks	Serum	Reduced levels of GOT, GPT, ALP, and LDH	[51]
					Liver	Decreased MDA production; and increased SOD, CAT, and GPx	
Broad-spectrum antibiotics (ABX)-induced tumour model	Male C57BL/6 mice	UPF	400 mg/kg/day	Oral gavage for 3 weeks	Tumour tissue	Reduced levels of CD31^+^, Bcl2; increased Bax level and CD8^+^ cells; and decreased CD4^+^ cells and IDO1 expression	[67]
					Faeces	Increased abundance of *Akkermansia*, *Bifidobacterium*, and *Lactobacillus*	
Cyclophosphamide-induced immunosuppression	Male SD rats	High molecular weight UPF	50, 100, and 200 mg/kg/day	Oral administration for 4 weeks	Faeces	Increased abundance of *Papillibacter cinnamivorans* and *Desulfomicrobium orale*; and reduced *Marvinbryantia formatexigens*	[68]
Ovalbumin-induced CARAS	Male BALB/c mice	Ethanol-extracted UPE	50, 100, and 200 mg/kg/day	Oral administration for 16 days	Serum	Attenuated IgE and IgG1 levels; and increased IgG2a	[71]
					Nasal lavage fluid	Increased expression of IFN-γ, SOD, and HO-1; reduced IL-4, IL-5, IL-13, and MDA; and enhanced ZO-1 and occludin	
					Bronchoalveolar lavage fluid	Decreased levels of IL-4, IL-5, IL-13, and MDA; and increased HO-1 and occludin production	
					Lung	Increased expression of occludin and ZO-1	
Salmonella typhimurium-induced inflammation	Male BALB/c mice	UPF	200 and 500 mg/kg/day	Oral administration for 21 days	Colon	Increased expression of occludin and claudin-1; reduced TNF-α, IKBα, p-IKBα, p65, and p-p65; elevated levels of CAT and SOD; and decreased MDA and iNOS	[72]
					Faeces	Reduced abundance of *Proteobacteria*, *Colidextribacter*, and *Oscillibacter*; increased *Parabacteroides*, *Lactobacillus*, *Akkermansia*, *Lachnospiraceae*_NK4A136 group and *Muribaculum*; and enhanced levels of acetate and butyrate	

**Table 3 marinedrugs-23-00350-t003:** In vitro neuroprotective activity of UPF.

Cell Line	Model	Compound	Tested Concentration	Effective Concentration	Activity	Reference
Human RPE cell line	HSV-1-induced Aβ production	HCl- extracted UPF	100 µg/mL	100 µg/mL	Inhibited NF-κB phosphorylation, IL-6 expression, and Aβ_42_ synthesis; and reduced DPPH scavenging and ROS production	[26]
Rat PC-12 cells	Aβ-induced neurotoxicity	UPF	3.125–100 µg/mL	3.125–100 µg/mL	Increased cell viability; reduced Aβ_1–42_ aggregation and cell apoptosis; and enhanced neurite outgrowth	[84]
PC12 cells	Aβ_25–35_ and d-Gal-induced neurotoxicity	Water-extracted UPF	100, 200, and 400 µg/mL	100, 200, and 400 µg/mL	Improved cell viability; prevented cell apoptosis; reduced levels of cleaved caspase-3, caspase-8, caspase-9, and cytochrome c; increased livin and X-linked apoptosis inhibitor protein expression; and elevated levels of SOD and GSH	[85]
Hypothalamic neurons (GT1-7 cells)	Tunicamycin- induced ER stress	Ethanol-extracted UPE	5–40 µg/mL	5 µg/mL	Increased cell viability; reduced expression of CHOP and ATF-6; decreased levels of cleaved-PARP and cleaved-caspase-3; and modulated AKT/mTOR signalling	[86]
BMDMs	LPS-induced macrophage inflammation	Low molecular weight UPF	0.0625, 0.125, 0.25, 0.5 mg/mL	0.125 mg/mL	Reduced CD86^+^ proportion; increased CD206^+^ proportion; regulated AMPK/mTOR pathway	[88]
Rat basal forebrain cholinergic neurons	Aβ-induced neurotoxicity	UPF	50 nM–1 µM	1 µM	Improved neuronal survival; inhibited ROS generation and PKC phosphorylation; and blocked cleavage of caspases 9 and 3	[90]
Rat hippocampal neurons	Hypoxia-mediated oxidative injury	Ethanol- extracted UPE	5, 15, 30 µg/mL	15 µg/mL	Reduced ROS formation; increased cell viability; and decreased cytotoxicity	[92]

**Table 4 marinedrugs-23-00350-t004:** In vivo neuroprotective activity of UPF.

Model	Animal	Compound	Dose	Treatment	Tissue	Result	Reference
HFD- induced obesity	Male and female C57BL/6J mice	UPF	400 mg/kg/day	Oral supplementation for 10 weeks	Hypothalamus	Reduced TNF-α, IL-1β, IL-6, and IFN-γ	[22]
					Nucleus accumbens	Suppressed Lrrk2, Wfs1, and Ngb	
SNL-induced neuropathic pain	Male SPF SD rats	UPF	15, 50, and 100 mg/kg/day	Intrathecal injection for 10 days	Lumbar spinal cord	Inhibited microglia and astrocyte activation; and reduced expression of GFAP and mac-1	[94]
					Spinal dorsal horn	Downregulated expression of TNF-α, IL-1β, and IL-6; and attenuated phosphorylation of ERK	
IRI-caused neurological impairment	Male SD rats	UPF	80 and 160 mg/kg/day	Intraperitoneal injection for 7 days	Ischemic brain	Reduced levels of TNF-α, IL-1β, IL-6, MPO, SOD, MDA, p-p53, p-p38, p-ERK, p-JNK, and Bax; and increased Bcl-2	[95]
D-Gal-induced AD model	Male ICR mice	UPF	50, 100, and 200 mg/kg/day	Oral administration for 21 days	Brain	Increased levels of Ach, ChAT, and GSH; reduced AChE activity; and decreased Aβ deposition	[85]
					Serum	Increased levels of SOD and GSH	
Controlled cortical impact-induced TBI	Male C57BL/6 mice	Low molecular weight UPF	10 and 50 mg/kg	Intraperitoneal injection	Brain	Decreased brain oedema and cell apoptosis; reduced generation of MDA, 4-HNE, and ROS; increased levels of CAT, SOD, and GPx; suppressed cytochrome c release; and upregulated Sirt3 expression	[96]
Aβ-induced AD model	Caenorhabditis elegans	UPF	50–500 ng/mL	Bath immersion method	Entire organism	Decreased Aβ deposition, aggregation, and fibrillization; increased expression of pbs-2 and pbs-5; and reduced ROS production	[97]

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
