# Peer review of "Anti-Inflammatory and Neuroprotective Effects of *Undaria pinnatifida* Fucoidan"

_marinedrugs, 2025, doi:10.3390/md23090350_

Round 1
Reviewer 1 Report
Comments and Suggestions for Authors
This manuscript provides a well-structured and comprehensive review of the anti-inflammatory and neuroprotective properties of fucoidan from Undaria pinnatifida (UPF). The review is timely and relevant, summarizing an impressive range of in vitro, in vivo, and limited clinical studies, and is well-referenced. However, the manuscript could be significantly improved by addressing several important gaps in the structural characterization of UPF, as these molecular features are known to critically influence its biological activity.
Major comments:
1. Lines 37–42 provide only a general overview of fucoidan’s backbone structure. It would greatly benefit readers if the authors included a short paragraph highlighting the unique structural characteristics of Undaria pinnatifida fucoidan (UPF), particularly in comparison to fucoidans from other brown algae. Key features to consider include its typical sulfation pattern, molecular weight range, degree of branching, and co-monomer composition. This addition would offer a stronger scientific foundation for interpreting the rest of the review.
2. Many studies cited in Sections 2 and 3 do not include crucial details about the UPF used, such as its source, molecular weight, degree of sulfation, monosaccharide composition, or extraction method. Since these parameters are well-documented determinants of fucoidan’s biological activity, their absence makes it difficult to evaluate or compare the reported anti-inflammatory and neuroprotective effects.
3. If the cited studies do not report structural parameters, the authors should explicitly acknowledge this as a limitation. Doing so would increase the transparency of the review and strengthen the validity of its conclusions. Without addressing these gaps, the overall synthesis may appear overly generalized.
4. Although some lines (e.g., 100–107, 188–195) briefly mention that low molecular weight or highly sulfated UPF fractions exhibit stronger bioactivities, the manuscript lacks a coherent discussion linking specific structural features to biological outcomes. A concise, dedicated paragraph discussing the structure–activity relationship would significantly enhance the interpretive depth of the review.
5. The manuscript is heavily text-based and would benefit from the inclusion of schematic figures to synthesize key mechanisms and findings. Suggested figures include:
- A schematic for Section 2.1 summarizing UPF’s in vitro anti-inflammatory mechanisms, such as cytokine inhibition, NF-κB/MAPK pathway modulation, and oxidative stress reduction.
- A schematic for Section 2.2 depicting in vivo effects, including systemic cytokine modulation, gut microbiota alterations, SCFA production, and neuroprotective outcomes.
Reviewer 2 Report
Comments and Suggestions for Authors
Cheng Yang et al have submitted the manuscript entitled 'Anti-Inflammatory and Neuroprotective Effects of Undaria pin-2 natifida Fucoidan'. After close evaluation of the manuscript I recommend revision according to the next comments and recommendations.
- The abstract is too general. No real results were provided. Please underline why this manuscript must be interesting for the scientific community.
- The aim looks as conclusion rather as aim. Please rephrase.
- The material and methods section is lacking. Please indicate which sources of information, key words, criteria for inclusion/exclusion, timeframes were used.
- Authors have referred many reviews. I would suggest to focused in experimental articles and provide own critical evaluation of published data.
- The efficacy of fucoidan is closely associated with its molecular weight, sulfates content, composition of monosaccharides, etc. I suggests to include this information in the Tables 1-4. The relationships must be discussed.
- Please include in the Tables information about positive control used. Please compare the efficacy of UPF with the efficacy of positive control.
- The discussion is weak. Please compare in discussion the efficacy of UPF with the efficacy of fucoidans from other brown algae. For example, In Vitro Anti-Inflammatory Activities of Fucoidans from Five Species of Brown Seaweeds was recently discussed. Notable, that purified fucoidan showed the most promising activity (IC50 = 0.20 mg/mL vs. IC50 = 0.37 mg/mL for diclofenac sodium). Similar relations were also observed in the membrane protection model.
- Fucoidans have shown efficacy after topical application as well. in one of the recent publication it was reported that topical application of the fucoidan-based cream dose-dependently inhibited carrageenan-induced edema and ameliorated mechanical allodynia in rats. The efficacy of the fucoidan-based cream at a high dose was comparable with the efficacy of diclofenac gel.
- AAntioxidant/ anti-radical activity is significantly contributed to the anti-inflammatory effects and closely related with the composition of fucoidan. In this respect I would suggests to compare the relation between biochemical composition, antiradical potential and human health risk forUndaria pinnatifida wuth biochemical composition, antiradical potential and human health risk other brown seaweeds sich as Fucus vesiculosus F, spiralis, F. distichus, Ascophyllum nodosum, etc.
- The pharmacokinetic of fucoidan is essential for understanding the anti-inflammatory activity of fucoidan. Although the pharmacokinetics of UPF have not yet been reported, the pharmacokinetics of fucoidans from other brown algae have been reported following oral and topical applications. Please discuss this aspect.
- aking in account importance of toxicity, please discuss toxicity of UPF in separate section.
- Are there any data on the clinical use of UPF?
- Please update the conclusion according above mentioned recommendations.
Round 2
Reviewer 1 Report
Comments and Suggestions for Authors
I appreciate the authors’ careful revisions and clear responses. The manuscript has been improved with more structural details, methodological transparency, and a useful discussion on limitations. The added section on structure–activity relationships and schematic figures make the paper stronger and easier to follow. Overall, the revisions are satisfactory, and the manuscript is suitable for publication
Reviewer 2 Report
Comments and Suggestions for Authors
Authors have significantly updated and revised the manuscript. Iy could be accepted in present revised form.